# Morphological Pattern of Building Clusters in Cold Regions: Evidence from Harbin

**DOI:** 10.3390/ijerph192417083

**Published:** 2022-12-19

**Authors:** Wente Pan, Shuqi Li, Yang Ye, Yuan Huang, Haocheng Liu, Hongxing Liu, Wenxuan Yu

**Affiliations:** Key Laboratory of Cold Region Urban and Rural Human Settlement Environment Science and Technology, Ministry of Industry and Information Technology, School of Architecture, Harbin Institute of Technology, Harbin 150000, China

**Keywords:** urban design, urban morphology, cold regions, pattern, building cluster, energy saving

## Abstract

The rapidly changing global conditions of the environment and climate have resulted in higher requirements for urban design. Significant annual temperature variations and large day/night temperature differences in cold-region cities leads to high energy consumption. Therefore, it is challenging to achieve low energy consumption in cold-region cities. Urban morphology focuses on the physical elements of urban areas, reflecting the relationship between the city and its environment and the city’s response to natural climatic conditions. Building clusters are common in cold regions due to the extreme climate. Thus, it is crucial to study the energy performance of cities by considering urban morphology. This study focuses on four morphological patterns of building clusters: point, linear, courtyard, and mixed patterns. A case study is conducted in Harbin, a cold-region city in China. Samples of the four morphological patterns are extracted, and GIS analysis and manual labeling are used to analyze the dominant morphological patterns of building clusters in cold regions. Average nearest-neighbor analysis is used to obtain quantitative results and determine the prevalence of different morphological patterns of building clusters in cold regions. This process can be used to determine the dominant patterns of urban building clusters and provide a scientific basis for selecting the morphological patterns of new building clusters in cold regions.

## 1. Introduction

### 1.1. Background

#### 1.1.1. Analysis of Cold-Region Building Clusters

Building clusters in cold regions represent an urban settlement that developed in cold and harsh climatic conditions [1]. They are a typical aspect of cold-region architecture, reflecting an adaptation to the special climate. The character of each element and the relations between elements are key factors in building cluster research. Most research on the morphology of cold-region building clusters has focused on the relationship between individual clusters and the environment. However, the morphological pattern relations between building clusters have not been investigated in detail, making it challenging for designers to understand the morphological relationship between new and existing buildings. Therefore, we introduce the concept of clustering to describe the distribution and pattern of building clusters with the same morphological characteristics in a city. The concept of clustering was first promoted as a statistical item for describing the autocorrelation of spatial data [2]. In spatial data analysis, clustering is used reflect the degree of aggregation of units with similar characteristics in the same cluster, i.e., the degree of interdependence between data at a certain location and at other locations [3]. Clustering analysis can describe the morphological correlations between cold-region building clusters, discover the implicit elements in the design, and provide designers with a solid foundation for urban design.

#### 1.1.2. Pattern

Pattern is a term commonly used in linguistics, mathematics, architecture, and other fields. Christopher Alexander introduced the concept of architectural pattern in his 1970 book *The Timeless Way of Building* [4]. He defined pattern as a way to describe good design practice or effective organization in a professional field. Pattern can also be regarded as several interconnected elements that express the wisdom of human beings through continuous practice, leading to holistic, positive, beautiful, variable, streamlined, and empirical results in a given domain [5]. A morphological pattern is a set of associations that build the various forms of a lexeme in linguistics. In this study, morphological pattern is regarded as an effective method in architectural and urban design to explore the meaning of basic form, as well as the relationship between clusters. 

### 1.2. Significance

The study of architectural clusters integrates architectural design and urban planning. Studies on architectural clusters fall into two categories. Architectural design studies focus on the spatial characteristics of architectural clusters from the perspective of historical heritage. In contrast, urban planning studies quantify architectural clusters from a macroscopic perspective. 

Due to the fossil energy crisis, carbon neutrality has become a crucial topic for the international community in recent years. A report published by the United Nations Intergovernmental Panel on Climate Change (IPCC) to assess global climate change stated that human use directly affects more than 70% of the global, ice-free land surface. The report also predicted future global temperature and sea level rises. Thus, the United Nations asked countries to take action. Rapid and far-reaching reforms in many fields are needed to achieve carbon neutrality, including energy, land, urban systems, and infrastructure [6]. 

The layout of cold-region building clusters is a passive measure that can improve energy efficiency in addition to building technology and is a response to urban environmental policies. On the one hand, building clusters are a congregation area for people living in cold areas through long-term natural selection. On the other hand, under extreme, changing, and complex environmental conditions, the whole is greater than the sum of individuals, i.e., building clusters better protect people from the cold. This study provides the following contributions to cold region architectural cluster research:(1)Expanding the design theory and research methods of cold-region building clusters. An analysis of urban morphology enables us to understand the patterns and the development of cold-region building clusters. In the past, the design of architectural clusters was a black-box process that only considered the macro perspective. This design approach reveals a lack of objectivity and rationality. This study combines data collection and analysis and uses a bottom-up approach to break through the limitations of designers’ perceptions to understand the reasons for cold-region building clusters. The results provide theoretical and technical support for the design of cold-region building clusters and expand the design theory and research methods of cold-region architecture.(2)Provide a reference for the architectural design of cold-region building clusters. We quantify the correlation between cluster types, summarize the distribution of different forms of architectural clusters in urban areas, explore the correlation between the morphological patterns of architectural clusters, and analyze the urban architectural design strategies in cold regions. The results provide new insights for morphological research and the creative practical applications of cold-region architectural clusters.

### 1.3. Literature Review 

The word “morphology” is derived from the Greek morphe (shape) and loqos (logic), meaning the logic of the composition of forms. Urban morphology refers to the study of urban form, the physical texture of cities, and the human, social-economic, and natural processes that shape their forms. In the field of urban design, urban morphology also refers to an analytical approach to the search for urban design principles [7,8]. Urban morphology focuses on human geography, historical heritage, and planning design as the main research perspectives and urban site selection, urban planning, and urban texture as the research objects to describe and investigate the principles of urban morphology, its evolution, and its influencing factors to guide urban design. 

Much of the research on early architectural clusters has focused on urban studies that considered architectural clusters as the basic units of urban planning and design. Examples include Thomas More’s [9] study of the relevance of the ideal urban spatial structural unit to the overall city in Utopia and Campanella’s consideration of streets and neighborhoods in the Sun City model. These early studies gave rise to the concept of the small unit in urban design and introduced the correlation between the small unit and the urban system. In his book *The Traffic and Settlements of People and their Dependence on Topography* published in 1841, Johann Georg Kohl [10] conducted the first systematic study of the morphological patterns of architectural clusters and a comparative study of different types of settlements, discussing the relationship between the distribution of architectural clusters and the topography and geographical environment. In 1963, Burton [11] introduced econometric methods into geographic research and created econometric geography, providing a concrete basis for the quantitative study of building clusters. In 1976, Bill Hillier and Julian Hanson [12] proposed the theory of spatial syntax based on graph theory to quantify the spatial correlations of building clusters. The connectivity, integration, and topological depth of spatial units in building clusters were calculated using axial and convex models. The quantitative representation of urban morphology enabled researchers to evaluate urban morphological characteristics, such as building layout, spatial indicators, and street functions. 

Since the beginning of the 19th century, many scholars have studied architectural clusters. Demangeon [13] established a relationship between rural culture and building clusters and classified traditional types of architectural clusters into four types: “long”, “block”, “star”, and “discrete”. The formation mechanisms of different types of building clusters were discussed in terms of natural and social motives. March and Martin at the University of Cambridge [14] proposed a concise model of urban areas and defined six geometric models of urban neighborhoods by studying the characteristics of European urban patterns. In response to outdoor climate, Olgyay [15] proposed five neighborhood morphology models, namely, east-west monolith, north-south monolith, east-west linear, north-south linear, and inner courtyard. In recent years, Jull [16] from the Polar Research Group of the University of Virginia proposed eight building cluster patterns for cold regions, and Merlier et al. [17] categorized them into five patterns in their study of wind environments. A literature review of studies on building cluster patterns indicates many classification methods for building clusters because the classification criteria and types depend on the research object and purpose. In addition, the same pattern may have different meanings in different research contexts. Therefore, the morphological patterns of building clusters are variable, and their classification and meaning must be defined in the urban context. 

Rode et al. [18] analyzed the relationship between urban form and energy consumption, stating that the choice of urban pattern is a subjective process. They sampled major residential building types in four cities with cold winters (Paris, London, Berlin, and Istanbul) to estimate thermal energy demand. It was concluded that compact neighborhoods with high building density had the highest neighborhood-scale thermal energy efficiency, whereas stand-alone dwellings had the lowest [19]. Leng et al. [20] quantified the influence of urban form factors on building energy consumption in cold regions using simulation and statistical analysis. They found that a higher floor area ratio and building site coverage resulted in higher heating energy consumption. Ahmadian et al. [21] classified urban building forms in London into four types: “pavilion”, “terrace”, “court”, and “tunnel-court”. “Court” has the lowest energy demand, and “pavilion” has the highest energy demand for similar geometric variables. Delmastro et al. [22] obtained similar results for a sample of buildings in Turin, Italy, and evaluated the planning level of the building clusters in the city, indicating that courtyard clusters were more energy efficient. The purpose of this paper is to classify the building cluster patterns in cold regions to determine the energy consumption of different patterns. The results can be used to incorporate cultural and energy aspects in future designs.

## 2. Materials and Methods

### 2.1. Main Methods and Tools

Clustering analysis is an analytical method for grouping and exploring the characteristics of different groups of research objects, i.e., exploring the relationship between elements and elements, elements and groups, and groups and groups. Clustering analysis is a common technique for spatial data processing and mining. It is primarily used for pattern recognition, information retrieval, image analysis, and other data analysis. Clustering analysis does not use a single algorithm but uses a workflow based on multiple algorithms.

There are significant differences in the clustering analysis of different research objects. Clustering analysis currently used in architectural research consists primarily of identifying groups with small distances between cluster members to detect dense regions in the data space or for determining the statistical distribution of various spatial elements. If architectural clusters are regarded as a language, we can develop a pattern language model based on a probability algorithm. In this language model, a city is regarded as a database, the blocks are considered the morphemes, the building clusters in the blocks are regarded as patterns, and the spatial relationships between the blocks are defined by the spatial correlation. In this case, the building clusters in cold regions can be represented as clusters of points aggregated by several similar units. The point clusters with spatial coordinates can be imported into a Geographic Information System (GIS) to analyze their aggregation characteristics. ArcGIS is used to count the morphemes in the pattern database, and an average nearest-neighbor analysis is performed to determine the aggregation characteristics of the point clusters.

Average nearest neighbor analysis (ANN) was proposed by Philip Clark and Francis Evans [21] in 1954 to determine the clustering characteristics of points. This method can be used to verify the development of building clusters [23,24]. The average nearest-neighbor ratio (the ratio of the average observed distance to the average expected distance) of individuals is calculated in this study. ANN, Ripley’s K function, local Moran’s I, and Getis-Ord Gi are popular spatial clustering assessment methods [25]. Unlike other clustering analysis methods, the ANN and Ripley’s K functions are intended more for point data, and the ANN method could compare the degree of aggregation of multiple data points for different data sets. Thus, it is more suitable for the analysis of building cluster morphology patterns in different scales [26].

### 2.2. Workflow

#### 2.2.1. Summary of Existing Cluster Patterns

The development of architectural clusters in cold-region cities has undergone a long evolutionary process, covering many historical periods. As a result of different environmental problems, various patterns of building clusters have developed. This study focuses on three patterns: point cluster, linear, and courtyard patterns. The three patterns have different characteristics and are attributed to the adaptation to the cold climate.

(1)Point cluster pattern to protect against the cold using a dense distribution. A point is the smallest unit of spatial elements, a basic element of recording spatial information in the human cognition of nature, and a basic morpheme in the language of cold-region building cluster patterns. A point cluster is a non-contiguous set of points in space, with a distance between the points. A point cluster is an abstracted unit in the cold-region building cluster. It can be decomposed into a single point, an aggregated point pattern, a random point pattern, and an evenly distributed point pattern (Figure 1).

(2)Linear pattern for cold protection according to the terrain. A line segment is another type of building cluster pattern. A river is a typical linear element in nature. Ancient settlements were built along rivers, forming a linear settlement pattern. The linear pattern has better environmental adaptability than the point cluster pattern. A linear pattern requires less land for settlement and avoids the negative space of the gable wall of the cold-region building. Linear buildings are conducive to forming spatial clusters and have high spatial levels. In terms of architectural form, a linear pattern is similar to the mountain and water elements in the environment, forming a unified picture (Figure 2).

(3)Courtyard pattern for cold protection by enclosing a space. The courtyard pattern is a typical cold-weather construction method. An area is enclosed by buildings to form a public external space and a private internal space, isolating the living area from the urban environment. The courtyard pattern effectively keeps out the cold in cold-region building clusters, and the surrounding buildings block unfavorable elements, such as cold winds and snowstorms, creating a relatively comfortable courtyard microenvironment. The courtyard pattern has a very long history in China, dating back to the Western Zhou Dynasty. It has been widely used in northern residential buildings and has evolved into various types according to different regional climatic characteristics (Figure 3).

(4)Mixed pattern: The block contains two or more of the basic patterns.

#### 2.2.2. Acquisition of Data on Morphological Patterns of Cold-Region Building Clusters

The adaptation of people to a cold climate has created a unique culture, resulting in cold-region architectural cluster patterns. The adaptation to the natural environment is a critical aspect of architectural cluster research. Cold-region architectural clusters have developed into multiple patterns over a long period to reflect different cold-adaptation strategies in a cold environment. The organization of building clusters is constantly updated, resulting in a specific syntax of pattern combinations to withstand the cold using different strategies.

In this paper, the block is regarded as the smallest unit to study the morphological patterns of architectural clusters, an indivisible area surrounded by streets or natural and artificial barriers.

The morphology of buildings in architectural clusters is diverse and complex. Therefore, we simplified the morphological patterns and its basic elements using the following criteria:(1)Point cluster pattern: The buildings in the block exhibit a point pattern, and the ratio of the longest side to the shortest side of a single building is less than 2:1.(2)Linear pattern: The buildings in the block have a linear pattern, and the ratio of the longest side to the shortest side of a single building is more than or equal to 2:1.(3)Courtyard pattern: The buildings in the block exhibit a courtyard pattern, consisting of a fully enclosed type, semi-open courtyard type, or combined type.(4)Mixed pattern: the blocks have characteristics of two or more of the above types.

The pattern identification of the block is carried out manually, based on the architectural form in its master plan. There are two steps to make the identification, the first is about the build types. When the ratio of the long side to the short side of the building is less than 2:1, it is judged as a point building; When the ratio of the long side to the short side of the building is more than or equal to 2:1, it is judged as a linear building; When a building presents a courtyard space, it is judged as a courtyard building. The second step is to identify the pattern of block according to the ratio of buildings in different types. When point buildings are more than half, the block could be judged as point cluster pattern; when linear buildings are more than half, the block could be judged as linear pattern; When courtyard buildings are more than half, the block could be judged as courtyard pattern; When multiple types of buildings coexist, the block could be judged as a mixed mode.

#### 2.2.3. Clustering Analysis of Morphological Patterns of Cold-Region Building Clusters

Average nearest-neighbor analysis is used to detect the global spatial clustering characteristics of elements. The distance between each point and its nearest point is calculated, and the average of the nearest point distance is determined to obtain the “average observed distance”. It is assumed that there is the same number of evenly distributed points in the space as in the sample, and the distance between the evenly distributed points is the average expected distance (Figure 4).

The average nearest neighbor ratio ANN is the ratio of the average observed distance to the expected average distance. When the ANN < 1, the distribution is clustered; if ANN > 1, the distribution is uniform or random distribution. The equations are as follows [27]:$$\mathrm{ANN}=\frac{{\;\bar{D}}_{0}}{{\;\bar{D}}_{E}}$$
where ANN is the average nearest neighbor rate; ${\bar{D}}_{0}$ is the average observed distance; ${\bar{D}}_{E}$ is the expected average distance.
$${\;\bar{D}}_{0}=\frac{\sum_{i=1}^{n}d_{i}}{n}$$
where $d_{i}$. is the distance between each element and its nearest element; n is the number of elements in the data set.
$${\;\bar{D}}_{E}=\frac{0.5}{\sqrt{\frac{n}{A}}}$$
where A is the area of the data set.

#### 2.2.4. Z-Score and *p*-Value

The z-score and *p*-value are used as statistical significance measures in the average nearest neighbor method [28]. The z-score, also called the standard score, is the difference between the average observed distance and the expected average distance of the spatial data divided by the standard deviation. The *p*-value represents the probability of the observed results having occurred by chance. The closer the *p*-value is to 1, the higher this probability is, and vice versa. Table 1 lists the significance levels of the *p*-value and z-score.

#### 2.2.5. Simulation

After calculating the ANN, the typical morphology mode in cold-region cities is simplified. Designbuilder is used to verify its energy consumption, proving that the continuation of the urban morphology, which means that increasing the clustering in cold-region cities will not only enhance the inheritance of the urban context, but also reduce energy consumption, which is conducive to green development. DesignBuilder is a mature and powerful software of EnergyPlus. DesignBuilder v7 (DesignBuilder Software Ltd, London, UK) includes the EnergyPlus v9.4 (Washington, WA, USA) engine with various performance and function enhancements. Therefore, this study uses Designbuilder to calculate the energy consumption of clusters, but the basic calculation of Designbuilder only considers the shelter between the buildings in the block, and does not adjust the indoor dynamic energy consumption according to the outdoor microclimate, especially the thermal environment. Therefore, ENVI-met (ENVI-met GmbH, Rhineland Pfalz, Germany) is supplemented to simulate the outdoor thermal environment in winter, assuming that the more clustered in typical morphology mode, the better the outdoor thermal environment, and thus a reduction in heating energy consumption.

### 2.3. Case Study and Data Collection

Harbin, a plain city developed since the beginning of the 20th century in northern China is used as a case study; seven areas are selected for analysis. Harbin is located in a high-latitude region, where the cold season lasts for more than 3 months, and the average temperature of the cold season is below 8 °C. Moreover, the selected cities have a long history of development and have adapted to the cold environment. Thus, Harbin is a representative cold-region city in China. In addition, this city has developed by self-organization [29], i.e., its growth was relatively spontaneous with little control. Thus, there is a close relationship between the building pattern and the cold environment.

The principles for the selection of the plots are as follows: (1) they underwent a long development; (2) they contain a similar number of blocks and have a similar area; (3) the area is surrounded by a railroad, highway, or natural elements; (4) they have a distinctive building pattern. After the case blocks were selected, the building outlines of the seven selected parcels in Harbin were obtained from the Open Map database and the World Imagery atlas. The selection of the research area is based on the consideration of the control variables, such as region size, region terrain, the years of construction, political background, main function, etc. Through the research on the historical development of the case city, as well as the field investigation and observation, 7 typical regions were chosen as the research object.

## 3. Results and Analysis

### 3.1. Block Model Type Analysis

ArcGIS was used to annotate the building cluster morphological patterns (Table 2). The numbers of different block types in the seven regions were determined, and the proportions of the four block pattern types were obtained, as shown in Figure 5 and Figure 6. All four patterns are present in the city. The point pattern accounts for 3% of all patterns, the mixed pattern account for 9%, the linear pattern accounts for 12%, and the courtyard pattern accounts for the largest proportion (76%), indicating that the courtyard pattern is the dominant building cluster pattern in this city.

### 3.2. Average Nearest-Neighbor Analysis

The GIS results indicate substantial differences in the clusters of different patterns, although the courtyard pattern is dominant in all seven regions. The comparison of the courtyard pattern clusters in region 5 and region 7 is shown in Figure 7. The courtyard pattern is more clustered in region 5 and more dispersed in region 7. Similarly, the linear pattern is more dispersed in region 3 and more clustered in region 4 (Figure 8).

Different block pattern types result in different clustering types in the same area of the same city, reflecting different layouts and affecting subsequent construction. Thus, it is necessary to quantify the clustering of different patterns in the areas.

Average nearest-neighbor analysis was performed in ArcGIS to obtain the ANN, z-score, and *p*-value for the four morphological patterns in each region. Figure 9 illustrates the results of the mean nearest-neighbor analysis for region 1, and Table 3 lists the summary of all the analysis results. In region 1, the ANN ratio for the courtyard pattern is 0.903, indicating that this pattern exhibits clustering. Its z-score is −2.02, and the *p*-value is 0.042, indicating significance and that the possibility of random occurrence of clustering is less than 5%. The ANN values of the point, linear, and mixed patterns in region 1 are greater than 1, suggesting that all three patterns are in a discrete state.

At the macro level, the results show that the number of clusters of the linear and courtyard patterns is relatively high, and that of the point and mixed patterns is relatively low. In other words, there are dominant morphological patterns in cities in cold regions, and the correlation between these morphological patterns is high.

At the meso level, the results show that 14 of the 28 morphological patterns are randomly distributed (50%), 9 have a homogeneous distribution (32.1%), and 5 exhibit clustering (17.9%) (Figure 10). The results suggest that most of the morphological patterns show dispersion, only some show clustering, and the courtyard pattern is dominant in these cities.

At the micro level, the results indicate that the courtyard pattern shows clustering in five of the seven samples, with an ANN ratio of 0.922, the lowest value among the four morphological patterns. The z-score and *p*-value are relatively high. The lower the ANN, the more pronounced the clustering is. The linear, point, and mixed patterns show dispersion or random distribution in the seven regions with average ANN values of 3.409, 1.167, and 1.185, respectively. These results indicate no clustering (Figure 11) and differences in the degree of clustering of different morphological patterns. The courtyard pattern shows substantial clustering, whereas the linear, point, and mixed patterns exhibit insignificant clustering. Therefore, the courtyard pattern has a higher morphological correlation, and the linear, point, and mixed patterns had lower morphological correlations.

### 3.3. Simulation Analysis

Since the typical morphology mode of cities in cold regions generally shows the clustered courtyard pattern, the mode is simplified into two typical models, the one is courtyard clustered with an ANN value of 0.92, and another is courtyard dispersed with an ANN value of 2.77 (Figure 12). Then, the energy consumption analysis is carried out. In terms of cooling energy consumption in summer, via Designbuilder, it is found that the annual cooling energy consumption of the clustered model of the courtyard is 61.63 kwh/m^2^, and the dispersed model is 63.87 kwh/m^2^. In the analysis of heating energy consumption in winter, considering the influence of the outdoor microclimate, the thermal environment of the clustering at the winter solstice is calculated by ENVI-met. It is found that the highest temperature of the clustered model is 20.7 °C, and the dispersed model is 20.2 °C (Figure 13). Therefore, if the courtyard pattern in cold regions shows clustered, the annual energy consumption of cooling and heating in the unit area is better than that in the dispersed one.

## 4. Discussion and Limitations

### 4.1. Clustering Characteristics of the Dominant Morphological Pattern

The statistical results demonstrated that the number of clusters was relatively high in the courtyard pattern and relatively low in the point, linear, and mixed patterns in Harbin, a city in a cold region in China. The point pattern had the fewest clusters in each area. A considerable difference was observed between the number of clusters in the courtyard pattern and the point, linear, and mixed patterns in the different regions, indicating the uniqueness of the courtyard pattern. The difference between the number of clusters in each pattern was relatively small in a few plots, but there was consistency in the number of clusters. Therefore, it was concluded that the courtyard pattern was the predominant morphological pattern in all regions of this cold-region city.

Additionally, a tendency of clustering in the plot center was discovered. The dominant morphological pattern of the site exhibited clustering. A polycentric distribution was observed in the city, with different clusters randomly dispersed in the areas. In other words, the influence of a pattern is stronger the closer a region is to the center of density, and other types of clusters occur near the edge.

The average nearest-neighbor analysis showed that not all morphological patterns displayed clustering. Only the predominant courtyard pattern exhibited clustering characteristics in most of the regions, with a high confidence level, and the courtyard pattern shows a dispersed distribution in a small number of plots. The point and mixed patterns showed dispersed distributions in most plots and random distributions in a few plots. The linear pattern primarily displayed random distributions and a few instances of dispersed distributions. None of the three morphological patterns showed clustering in any plots.

In addition, the chosen plots developed over a long period due to the influence of the cold climate. The point cluster pattern primarily showed a dispersed distribution. This distribution results in a lower resistance to the cold climate. In contrast, uniformly distributed building clusters result in a more comfortable indoor environment in winter. Courtyards create outdoor activity spaces that are more comfortable in freezing winter climates. Each building cluster generates heat and protects from the wind due to the clustering effect. Thus, building clusters result in heat islands, conserving energy and improving the indoor environment.

In summary, the results of this study indicate that not all building clusters in the city exhibited the same degree of clustering. The courtyard pattern showed considerable clustering, whereas the other morphological patterns did not exhibit significant clustering. In a self-organized cold-region city, urban morphology was largely restricted by the climate conditions and building clusters increase the resistance to cold and improve the built environment. According to the conclusion that the energy consumption of high-density urban form and enclosed architectural form was low during building operations mentioned in previous studies, we can conclude that selecting the appropriate pattern in this way is beneficial not only in energy consumption, but also in continuation of the context.

### 4.2. Inspiration and Limitations

The results of this study found that the courtyard pattern accounted for the highest proportion in cold regions and showed a high degree of clustering. A classification of typical urban forms and the average nearest-neighbor analysis are objective methods to evaluate building cluster design. Additionally, we used information technology to prevent the contradiction between the traditional top-down design approach and the self-organization of building clusters. We analyzed the differences in the morphological patterns of building clusters. Against the background of environmental and climate issues, this study represents a rational analysis of the impact of building clustering on energy consumption in cold regions from a macro perspective. We proposed a scientific method and theoretical basis for the design of cold-region building clusters adapted to the lives of people in cold regions while achieving energy savings.

This study has some limitations. We only focused on building clusters in Harbin, a typical cold-region city in northern China. Different countries and regions have different cultural backgrounds and natural environments. Thus, different results may be obtained in other cities. Designers should adjust their designs to local conditions and considers the urban environment and background. They should conduct specific analyses outlined in this paper, combining data analysis with traditional design using data models and analysis tools to improve the design of cold-region building clusters while meeting various needs to minimize energy consumption.

## 5. Conclusions

Architectural clusters in cold regions and their effects on energy consumption have not been investigated sufficiently. In many cases, architectural patterns common in warmer regions are used in the design of cold-region cities, resulting in high energy consumption. Rational and green design have become new themes in the information age. We classified cold-region morphological patterns of building clusters into point cluster, linear, courtyard, and mixed patterns and analyzed them in ArcGIS. Average nearest-neighbor analysis was conducted to quantify the morphological relationships between building clusters. It was found that the morphological patterns of building clusters were significantly different for different patterns. The courtyard pattern exhibited the most clustering, followed by the point, linear, and mixed patterns, and was the most energy-efficient pattern. Future studies of cold-region cities should increase the sample size and investigate the development of clusters in different cities. The proposed methodology can be used by researchers and designers to establish the morphological correlation between architectural cluster systems to design urban morphology in cold regions.

## Figures and Tables

**Figure 1 ijerph-19-17083-f001:**
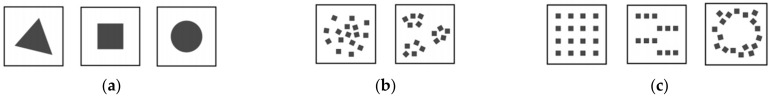
Point cluster pattern. (**a**) single point; (**b**) aggregated points; (**c**) evenly distributed points.

**Figure 2 ijerph-19-17083-f002:**
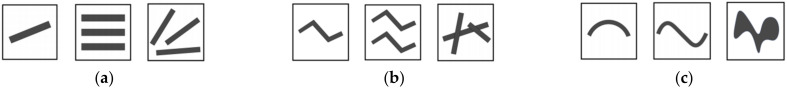
Linear pattern. (**a**) Long segment; (**b**) Folded line; (**c**) Curve.

**Figure 3 ijerph-19-17083-f003:**
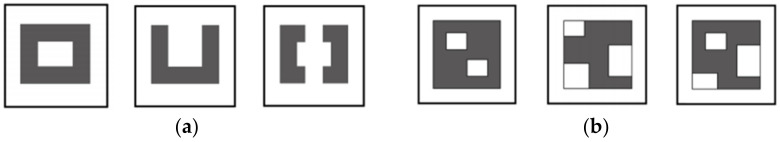
Courtyard pattern. (**a**) single courtyard; (**b**) Multilevel courtyard.

**Figure 4 ijerph-19-17083-f004:**
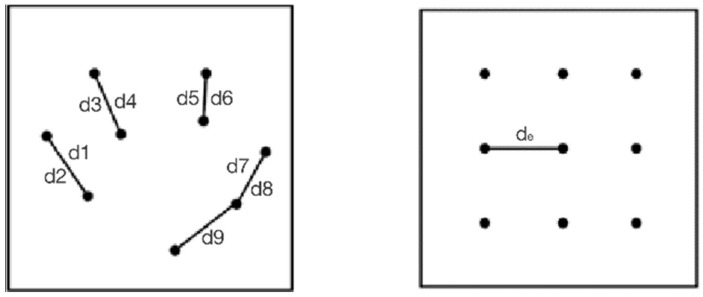
Average observed distance vs. expected average distance.

**Figure 5 ijerph-19-17083-f005:**
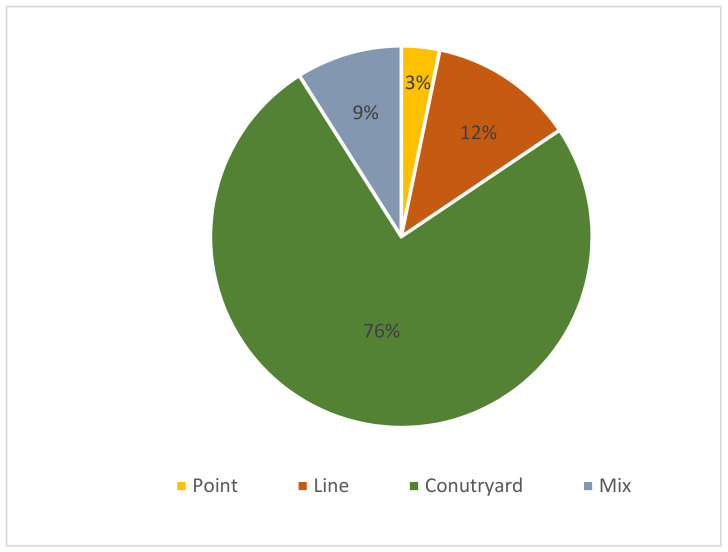
Proportions of block patterns in the city.

**Figure 6 ijerph-19-17083-f006:**
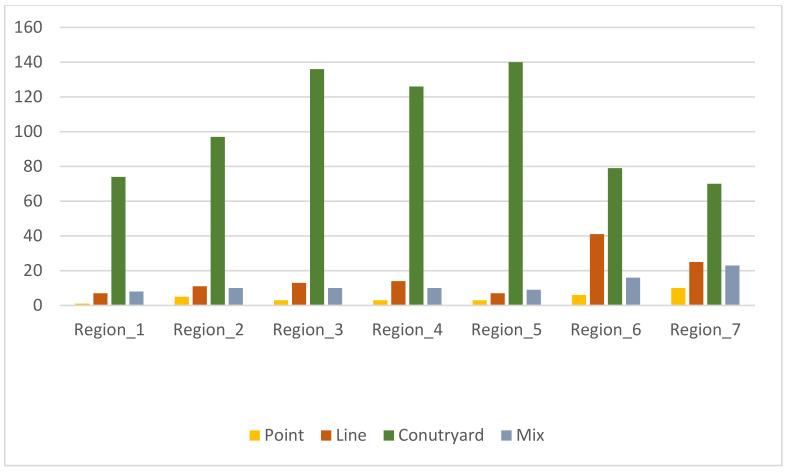
Number of 4 block patterns in the study areas.

**Figure 7 ijerph-19-17083-f007:**
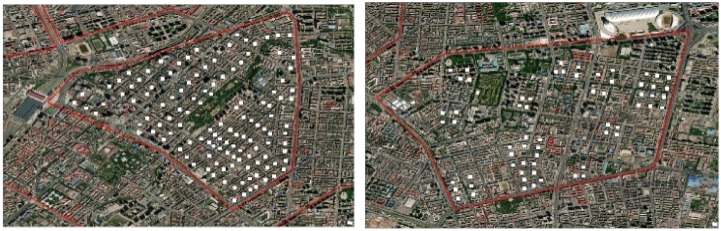
Comparison of the courtyard pattern clusters in region 5 and region 7.

**Figure 8 ijerph-19-17083-f008:**
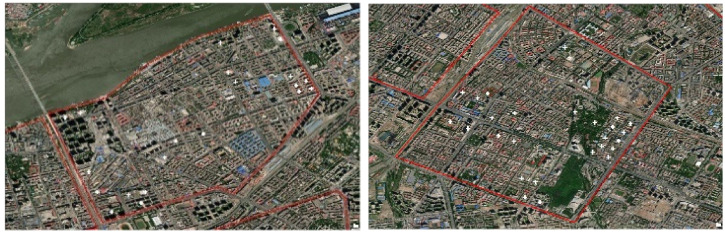
Comparison of the linear pattern clusters in region 3 and region 4.

**Figure 9 ijerph-19-17083-f009:**
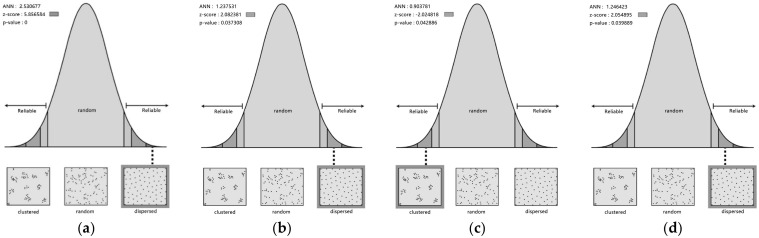
Results of average nearest-neighbor analysis in region 1. (**a**) point; (**b**) line; (**c**) courtyard; (**d**) mix.

**Figure 10 ijerph-19-17083-f010:**
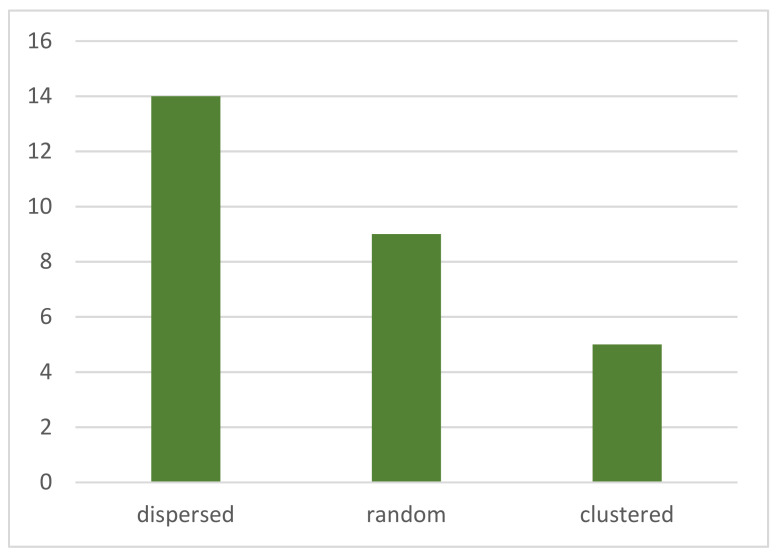
Types of morphological patterns.

**Figure 11 ijerph-19-17083-f011:**
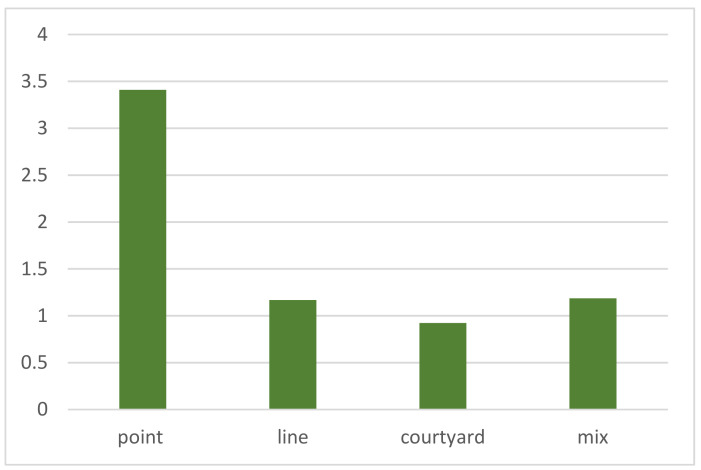
Average ANN value of the morphological patterns in seven regions.

**Figure 12 ijerph-19-17083-f012:**
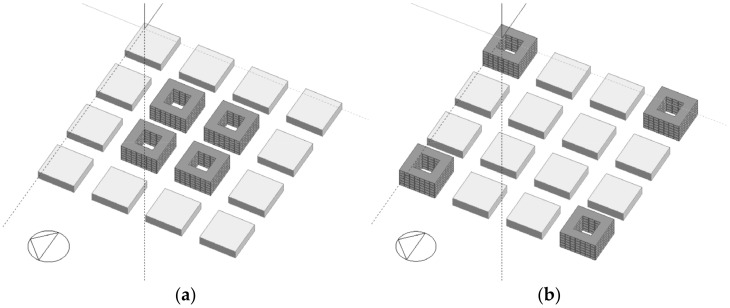
Simplified morphological clusters. (**a**) clustered; (**b**) dispersed.

**Figure 13 ijerph-19-17083-f013:**
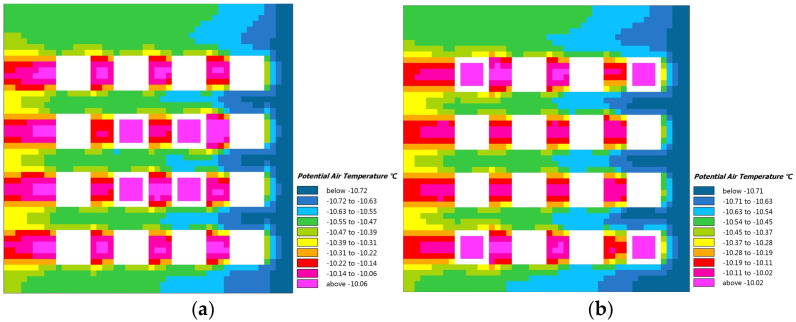
Thermal environment in winter. (**a**) clustered; (**b**) dispersed.

**Table 1 ijerph-19-17083-t001:** Significance levels of the z-score and *p*-value.

Z Score (Standard Deviation)	*p*-Value (Probability)	Confidence Level
<−1.65 or >+1.65	<0.10	0.9
<−1.96 or >+1.96	<0.05	0.95
<−2.58 or >+2.58	<0.01	0.99

**Table 2 ijerph-19-17083-t002:** The morphological patterns of cold-region building clusters.

Region	Region_1	Region_2	Region_3	Region_4	Region_5	Region_6	Region_7
Satellite image	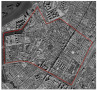	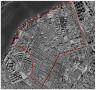	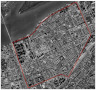	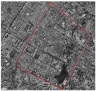	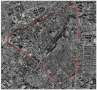	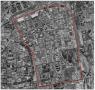	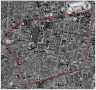
Dataset	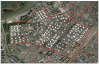	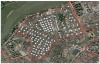	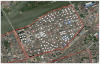	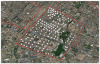	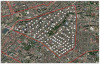	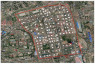	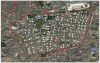
Point	1	5	3	3	3	6	10
Line	7	11	13	14	7	41	25
Courtyard	74	97	136	126	140	79	70
Mix	8	10	10	10	9	16	23

**Table 3 ijerph-19-17083-t003:** Results of the average neighbor analysis for different morphological patterns in different regions.

	Region_1	Region_2	Region_3	Region_4	Region_5	Region_6	Region_7
point cluster	dispersed	dispersed	random	Dispersed	dispersed	dispersed	random
ANN	2.530677	1.69431	1.021969	1.302401	14.606506	1.729342	0.976849
z-score	5.856584	3.514257	0.111198	1.918713	45.085630	3.691568	−0.153422
*p*-value	0	0.000441	0.911459	0.05502	0	0.000223	0.878066
linear cluster	dispersed	dispersed	random	Random	dispersed	random	random
ANN	1.237531	1.298635	0.951383	1.154374	1.504588	0.986570	1.034717
z-score	2.082381	2.137645	−0.426214	1.505882	2.553974	−0.170430	0.351436
*p*-value	0.037308	0.032546	0.669952	0.132097	0.010650	0.864672	0.725261
courtyard cluster	clustered	dispersed	clustered	Clustered	dispersed	clustered	clustered
ANN	0.903781	1.022213	0.906057	0.906820	1.114977	0.733709	0.872878
z-score	−2.024818	0.407599	−1.714423	−1.672223	2.459217	−4.107178	−1.930283
*p*-value	0.042886	0.683568	0.086451	0.094480	0.013924	0.000040	0.053572
mixed cluster	dispersed	random	random	Dispersed	dispersed	random	dispersed
ANN	1.246423	1.124625	1.081049	1.181489	1.340857	1.152715	1.170588
z-score	2.054895	1.327442	1.107301	1.994521	1.956252	1.573301	1.985090
*p*-value	0.039889	0.184363	0.268164	0.046095	0.050435	0.115649	0.047134

## Data Availability

Not applicable.

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
