# Peer review of "Morphological Pattern of Building Clusters in Cold Regions: Evidence from Harbin"

_ijerph, 2022, doi:10.3390/ijerph192417083_

Round 1
Reviewer 1 Report
The basic concept of spatial pattern identification is fine and work has scientific relevancy, but the article has more problematic aspects
1. I don't know the authors' professional background, but the need to define terms like pattern (without the adjective spatial) and clustering (moreover, to cite Prof. Tobler's 50 year old work in this context is odd (not irrelevant, but there is no particular reason to link spatial autocorrelation to him), to introduce arcgis and other similar references in the text - I assume this is neither geoinformation science nor settlement geography. There is a plethora of studies in the aforementioned disciplines analysing both settlement patterning and environmental influences on settlement structure.
2. If the title and references in the text associate spatial pattern with cold climate, the question is to what extent the configuration of settlements itself is determinant, because one would need to distinguish environmental influences from other factors (e.g. socio-economic). Even if we restrict ourselves to environmental influences - terrain is probably a more significant factor for settlement structure than climate
3. Lack of comparison - both with similar studies and identified patterns in other climates
4. Not clear to me the selection of 7 regions - why the number of 7, how were they defined - the characteristics provided are somewhat vague and unjustified, determined manually by looking?
5. The choice of clustering method is insufficiently justified
6. Identification of patterns by the chosen method is nowhere described
Author Response
Dear Reviewer
Thank you very much for your comments, which have brought important help to the revision of the article. In response to the comments, the following amendments are made, you can also see the attachment:
- I don't know the authors' professional background, but the need to define terms like pattern (without the adjective spatial) and clustering (moreover, to cite Prof. Tobler's 50 year old work in this context is odd (not irrelevant, but there is no particular reason to link spatial autocorrelation to him), to introduce arcgis and other similar references in the text - I assume this is neither geoinformation science nor settlement geography. There is a plethora of studies in the aforementioned disciplines analysing both settlement patterning and environmental influences on settlement structure.
Answer: Accept, in the revised manuscript, the discussions on the definition of “clustering” and “pattern” are rewritten, see 1.1Line 38-42 and 55-58. For the plethora discussion on the background of GIS has been deleted.
- If the title and references in the text associate spatial pattern with cold climate, the question is to what extent the configuration of settlements itself is determinant, because one would need to distinguish environmental influences from other factors (e.g. socio-economic). Even if we restrict ourselves to environmental influences - terrain is probably a more significant factor for settlement structure than climate
Answer: The settlement structure is not determined by a single factor. However, in the process of selecting samples, we controlled the macro variables and selected the plain cities in Northeast China. In addition, we found that the courtyard mode accounted for a large proportion in the experiment. Combining the conclusion that the energy consumption of high density urban form and enclosed architectural form was low during building operations mentioned in previous studies, we can find that the formation of the cold city morphology was largely restricted by the climate conditions, and a low energy consumption method was chosen. This content has been added in the revised manuscript, please see line 304 to 305.
- Lack of comparison - both with similar studies and identified patterns in other climates
Answer: The patterns should be the same in different climatic regions, but the proportions are different.
- Not clear to me the selection of 7 regions - why the number of 7, how were they defined - the characteristics provided are somewhat vague and unjustified, determined manually by looking?
Answer: The selection of the research area is based on the consideration of the control variables, such as region size, region terrain, the years of construction, political background, main function, ect. Through the research on the historical development of the case city, as well as the field investigation and observation, typical regions were chosen as the research object. There is no special setting for the number of selected regions, 7 areas with similar variables were found in the case city. This content has been added in the revised manuscript, please see line 318 to 322.
- The choice of clustering method is insufficiently justified
Answer: Accept. ANN, Ripley’s K function, local Moran’s I and Getis-Ord Gi are popular spatial clustering assessment methods. Unlike other clustering analysis methods, the ANN and Ripley’s K functions are more intended for point data, and the ANN method could compare the degree of aggregation of multiple data points for different data sets. Thus, it is more suitable for the analysis of building cluster morphology patterns with different scales. The Basis of choosing ANN has been illustrated on Line 199 to 204 in the revised manuscript.
- Identification of patterns by the chosen method is nowhere described
Answer: Accept, the chosen method need to be added. The pattern identification of the block is carried out manually, based on the architectural form in its master plan. There are two steps to make the identification, first is about the build types. When the ratio of the long side to the short side of the building is less than 2:1, it is judged as a point building; When the ratio of the long side to the short side of the building is more than or equal to 2:1, it is judged as a linear building; When a building presents a courtyard space, judge it as the courtyard building. The second step is to identify the pattern of block according to ratio of buildings in different types. When point buildings are more than half, the block could be judged as point cluster pattern; when linear buildings are more than half, the block could be judged as linear pattern; When courtyard buildings are more than half, the block could be judged as courtyard pattern; When multiple types of buildings coexist, the block could be judged as a mixed mode. This content has been added in the revised manuscript, please see line 264 to 274.
Reviewer 2 Report
This manuscript provides some interesting information, however revision is recommended before it may be considered, namely:
Line 57-64. Place reference to the cited document.
Line 242. in that line of text you mention "...of a single building is <2.". What does the 2 mean?, 2 m, 2 km, 2 milles?
Line 244. What does the 2 mean?, 2 m, 2 km, 2 milles?
Line 395. How are energy savings achieved? How are those savings measured? On what basis do you conclude that there are energy savings?. A detailed analysis of energy consumption is necessary to be able to conclude that with a defined pattern of construction, savings in energy consumption are achieved.
Author Response
Dear Reviewer
Thank you very much for your comments, which have brought important help to the revision of the article. In response to the comments, the following amendments are made, you can also see the attachment:
- Line 57-64. Place reference to the cited document.
Answer: We supplemented the source of the report. IPCC, 2019: Climate Change and Land. https://www.ipcc.ch/site/assets/uploads/sites/4/2021/02/210202-IPCCJ7230-SRCCL-Complete-BOOK-HRES.pdf (27th No-vember 2022)
- Line 242. in that line of text you mention "...of a single building is <2.". What does the 2 mean? 2 m, 2 km, 2 milles? Line 244. What does the 2 mean? 2 m, 2 km, 2 milles?
Answer: “2” of the above two questions refers to “ratio”. In line 258, 260, the text is modified to “the ratio of the longest side to the shortest side of a single building is less than 2:1” and “more than or equal to 2:1”.
- Line 395. How are energy savings achieved? How are those savings measured? On what basis do you conclude that there are energy savings? A detailed analysis of energy consumption is necessary to be able to conclude that with a defined pattern of construction, savings in energy consumption are achieved.
Answer: We found that the courtyard mode accounted for a large proportion in the experiment. Combining the conclusion that the energy consumption of high density urban form and enclosed architectural form was low during building operations mentioned in previous studies, we can find that the formation of the cold city morphology was largely restricted by the climate conditions, and a low energy consumption method was chosen. Thus, we concluded that according to the workflow of this study, we can choose the most appropriate pattern, which is usually the most energy-saving in cold regions. This method is beneficial not only in energy consumption, but also in continuation of the context. This content has been added in the revised manuscript, please see line 411 to 417.
Round 2
Reviewer 2 Report
In discussion and limitations part, you mention the following: "We proposed a scientific method and theoretical basis for the design of cold-region building clusters adapted to the lives of people in cold regions while achieving energy savings." (lines 403-405). How do you come to that conclusion?. How are energy savings achieved?. How are those savings measured? On what basis do you conclude that there are energy savings?. A detailed analysis of energy consumption is necessary to be able to conclude that with a defined pattern of construction, savings in energy consumption are achieved.
In the previous review, I asked the same thing and them didn't answer anything about it.
Author Response
Answer: Accept. After calculating the ANN, the typical morphology mode in cold-region cities is simplified. Designbuilder is used to verify its energy consumption, proving that the continuation of the urban morphology, which means that increasing the clustering in cold-region cities will not only enhance the inheritance of the urban context, but also reduce energy consumption, which is conducive to the green development. DesignBuilder is a mature and powerful software of EnergyPlus. DesignBuilder v7 includes the EnergyPlus v9.4 engine with various performance and function enhancements. Therefore, this study uses Designbuilder to calculate the energy consumption of clusters, but the basic calculation of Designbuilder only considers the shelter between the buildings in the block, and does not adjust the indoor dynamic energy consumption according to the outdoor microclimate, especially the thermal environment. Therefore, envi-met is supplemented to simulate the outdoor thermal environment in winter, assuming that the more the clustered of typical morphology mode, the better the outdoor thermal environment, and thus reduce the heating energy consumption.Since the typical morphology mode of cities in cold regions generally shows the clustered of courtyard pattern, the mode is simplified into two typical models, the one is courtyard clustered with ANN value of 0.92, and another is courtyard dispersed with ANN value of 2.77. Then, the energy consumption analysis is carried out. In terms of cooling energy consumption in summer, via Designbuilder, it is found that the annual cooling energy consumption of the clustered model of the courtyard is 61.63 kwh/m2, and the dispersed model is 63.87 kwh/m2. In the analysis of heating energy consumption in winter, considering the influence of outdoor microclimate, the thermal environment of the clustering at the winter solstice is calculated by envi-met. It is found that the highest temperature of the clustered model is 20.7℃, and the dis-persed model is 20.2℃. Therefore, if the courtyard pattern in cold regions shows clustered, the annual energy consumption of cooling and heating in unit area is better than that in dispersed one.
This part of simulation has been illustrated on 2.2.5.simulation (Line 303 to 317), and 3.3 Simulation analysis (Line 390 to 402) in the revised manuscript.